# New Piperazine Derivatives of 6-Acetyl-7-hydroxy-4-methylcoumarin as 5-HT_1A_ Receptor Agents

**DOI:** 10.3390/ijms24032779

**Published:** 2023-02-01

**Authors:** Kinga Ostrowska, Anna Leśniak, Weronika Gryczka, Łukasz Dobrzycki, Magdalena Bujalska-Zadrożny, Bartosz Trzaskowski

**Affiliations:** 1Department of Organic and Physical Chemistry, Faculty of Pharmacy, Medical University of Warsaw, 1 Banacha Str., 02-097 Warsaw, Poland; 2Centre for Preclinical Research and Technology, Department of Pharmacodynamics, Faculty of Pharmacy, Medical University of Warsaw, 1 Banacha Str., 02-097 Warsaw, Poland; 3Crystallochemistry Laboratory, Chemistry Department, Warsaw University, 1 Pasteura Str., 02-093 Warsaw, Poland; 4Centre of New Technologies, University of Warsaw, 2C Banacha Str., 02-097 Warsaw, Poland

**Keywords:** 6-acetyl-7-hydroxy-4-methylcoumarin, piperazine, serotonin receptors, CNS activity, crystal structure

## Abstract

A series of 15 new derivatives of 6-acetyl-7-hydroxy-4-methylcoumarin containing a piperazine group were designed with the help of computational methods and were synthesized to study their affinity for the serotonin 5-HT_1A_ and 5-HT_2A_ receptors. Among them, 6-acetyl-7-{4-[4-(3-bromophenyl)piperazin-1-yl]butoxy}-4-methylchromen-2-one (**4**) and 6-acetyl-7-{4-[4-(2-chlorophenyl)piperazin-1-yl]butoxy}-4-methylchromen-2-one (**7**) exhibited excellent activity for 5-HT_1A_ receptors with Ki values 0.78 (0.4–1.4) nM and 0.57 (0.2–1.3) nM, respectively, comparable to the Ki values of 8-OH-DPAT (0.25 (0.097–0.66) nM). The equilibrium dissociation constant values of the tested compounds showed differential intrinsic activities of the agonist and antagonist modes.

## 1. Introduction

5-HT receptors belong to the group of G-protein-coupled membrane receptors located on the cell membrane of neurons and selected other cells, including smooth muscle, pancreatic β-cells, hepatocytes and adipocytes. 5-HT receptors mediate the action of serotonin both in the central nervous system and the periphery nerves [1,2,3,4]. They are also an important target for a variety of drugs for the treatment of anorexia, schizophrenia, psychosis or depression [1,2,3,4]. 

*N*-arylpiperazine-containing ligands are one of the families of chemical compounds known to strongly interact with serotonin receptors [5,6]. This large group owes its biological properties to the presence of a highly basic nitrogen atom of the piperazine. This atom can form strong interactions with the acidic amino acids in the GPCR transmembrane domain of proteins [7]. Coumarin–piperazine ligands are also known for their cytotoxic activity, acting as antibacterial and antifungal agents, acetylcholinesterase inhibitors and as dual serotonin and dopamine receptor agents with antipsychotic and antiparkinsonian properties [8]. These compounds, often bearing arylpiperazines linked to a coumarin system via an alkyl linker, can modulate central nervous system (CNS) affective function by targeting serotoninergic, dopaminergic and adrenergic receptors. From a medical point of view, the importance of finding new biologically active compounds lies in the fact that serotonin and dopamine receptors are involved in the pathomechanisms of many psychiatric and neurological disorders, such as schizophrenia, depression, epilepsy or Alzheimer’s and Parkinson’s diseases [8].

In this work, as a continuation of our previous research, 6-acetyl-7-hyroxy-4-methylcoumarin was used as the lead compound for further structural modifications [9,10,11,12,13,14,15,16]. Recently, we synthesized a series of aryl/heteroarylpiperazinyl derivatives of 8-acetyl-7-hydroxy-4-methylcoumarin and evaluated their antidepressant-like activity [9,10]. These compounds showed very high affinity for serotonin 5-HT_1A_ receptors (Figure 1). The acetyl group in position 8 of the coumarin ring of these compounds increased the affinity for 5-HT_1A_ and 5-HT_2A_ receptors compared to their derivative bearing no acetyl group in position 8 [9]. Here, we decided to attach the acetyl group with its ability to form hydrogen bonds with residues in the 5-HT_1A_ binding pocket, at the C-6 position of the coumarin ring. We designed a synthetic strategy for target compounds with different aryl and heteroarylpiperazine moieties and harboring a four-carbon linker between the coumarin and piperazine ring, which according to earlier studies, gave the most favorable binding features [9,10]. Following the design of new coumarin derivatives, we used molecular docking to multiple crystal structures of 5-HT_1A_ and 5-HT_2A_ receptors to estimate their affinities, followed by microwave-assisted protocols, which were used to synthesize all new compounds. Upon the successful synthesis all compounds, they have been evaluated for their binding affinities for 5-HT_1A_ and 5-HT_2A_ receptors and their agonistic/antagonistic properties at the 5-HT_1A_ receptor.

## 2. Results and Discussion

### 2.1. Chemistry

The starting coumarin 6-acetyl-7-hydroxy-4-methylchromen-2-one was resynthesized according to previously published studies [17]. The planned compounds were obtained in two steps. In the first step, 6-acetyl-7-(4-bromobutoxy)-4-methylchromen-2-one (**A**) was prepared by alkylation of the phenolic group with dibromobutane in acetonitrile in the presence of potassium iodide and potassium carbonate (Figure 1). In the second step, the final compounds were obtained according to the previously published study [16]. Microwave irradiation was used in order to increase yield and to reduce the reaction time. The synthesis of compounds **1**–**15** was carried out by reacting the 6-acetyl-7-(4-bromobutoxy)-4-methylchromen-2-one (**A**) with appropriate arylpiperazine: 4-(2-methoxyphenyl)piperazine, 4-(3-methoxyphenyl)piperazine, 4-(2-bromophenyl)piperazine, 4-(3-bromophenyl)piperazine, 4-(4-bromophenyl)piperazine 4-(2-fluorophenyl)piperazine, 4-(2-chlorophenyl)piperazine, 4-(2-cyanophenyl)piperazine, 4-(2,3-dichlorophenyl)piperazine, 4-(3,5-dimethylphenyl)piperazine, 4-(2,5-dime thylphenyl)piperazine, 4-(4-nitrophenyl)piperazine, morpholine, 4-(pyrid-4-yl)pipera zine and 4-(pyrazin-2-yl)piperazine, in acetonitrile and in the presence of potassium iodide and potassium carbonate. The TLC method was used to monitor the progress of the reaction (silica gel plates, eluent: CHCl_3_: MeOH; 10: 0.25). We also used microwave irradiation in this step and column chromatography (silica gel, CHCl_3_:MeOH (100:1) as the eluent) to purify the final compounds **1**–**15**. The final product yields were in the 31–95% range. All compounds were characterized using standard methods, ^1^H NMR, ^13^C NMR spectroscopy, and HRMS spectrometry. All NMR spectra are presented in the Appendix A. To complete the structural characterization of compounds, we also report results of X-ray crystallographic studies for 6-acetyl-7-{4-[4-(3-methoxyphenyl)piperazin-1-yl]butoxy}-4-methylchromen-2-one (**2**).

### 2.2. X-ray Crystallography

The crystal data and structure refinement parameters for 6-acetyl-7-{4-[4-(3-methoxyphenyl)piperazin-1-yl]butoxy}-4-methylchromen-2-one (**2**) are collected in Table 1. Compound **2** crystallizes in *P*2_1_/*c* space group. The structure of the crystal measured at 130 K is fully ordered, which is visible in Figure 2, displaying the asymmetric unit with anisotropic displacement parameters of all non-H atoms at a 50% probability level. 

Because of the lack of strong hydrogen bond donors, the structure is supported by weak interactions only. However, such interactions slightly affect the geometry of the coumarin molecules, which deviate somewhat from planarity. The distance between the mean plane fitted to the coumarin C5–C10 atoms and O2 moiety is 0.15 Å. In addition, the acetyl group is slightly rotated with respect to the C5–C10 benzene moiety with the interplanar angle close to 15°. In the crystal lattice, the coumarin fragments are engaged in parallel π-π stacking, with the closest distances between the C5–C10 benzene plane and above/below coumarin moiety C atoms being ca. 3.52 Å. These stacks of moieties are parallel to the [001] direction. The side chains containing the piperazine unit are grouped in the central part of the unit cells, as presented in Figure 3. Here, some weak C H…N interactions (H…N distance of 2.64 Å) involving neighboring piperazine moieties can be observed.

### 2.3. Computational Studies

#### 2.3.1. ADME Properties

The major predicted ADME properties of all compounds studied are presented in Table 2. Some of the newly synthesized compounds exceed the desired common limit of 500 Da for systems with good oral bioavailability [18], but only slightly, and they all fall within the modern limit of <700 Da [19]. Compared to aripiprazole, there is a higher number of possible hydrogen bond acceptors due to the presence of one or two more oxygen atoms in the coumarin scaffold. As a result of the same structural feature, all investigated compounds were predicted to be more soluble in water. Finally, in all cases apart from **14**, the nitrogen atom of the piperazine part of ligands is predicted to be basic and protonated, as in the case of aripiprazole (although, the expected accuracy of pKa estimate is of around 0.7–1.0 pH units).

#### 2.3.2. Molecular Docking

The results of the Ki estimates obtained from the local search are presented in Table 3. For both the 5-HT_1A_ and 5-HT_2A_ receptors, we were able to find several compounds with expected higher affinity for these receptors than aripiprazole. However, in most cases, the difference between compounds with the highest estimated affinity and aripiprazole is below 1 kcal/mol or one order of magnitude in the Ki estimates, which is likely below the accuracy of the docking method [20]; on the other hand, since all poses in the local search are very similar, it is likely that error cancelation occurs. This is also clear when comparing the computational Ki estimate of aripiprazole in the pose from the crystal structure (45.6 nM) with the experimental value of 1.5–5.6 nM [21,22,23]. 

Comparing the results of the local search with the experimental Ki values presented in the next part of this study, the agreement between those two methods is only average. For example, compounds **12**–**15** were correctly predicted to have low affinities for the 5-HT_1A_ receptor. Additionally, compounds **1**–**11** were predicted to have Ki values between 20 and 70 nM for the 5-HT_1A_ receptor, which is in good agreement with the experimental values of 0.5–22 nM given the expected accuracy of the docking approach. To improve the computational results, after performing the biological affinity evaluation, we also conducted flexible docking with the Lamarckian genetic algorithm to multiple crystal structures of the 5-HT_1A_ receptor (see Table 3). Here, the results were improved, and in many cases, the obtained affinity estimates were very close to the experimental values. On the other hand, for selected systems with relatively low affinities (e.g., **12**), flexible docking predicts very low Ki values, which are clearly in disagreement with experimental data. In these cases, the poses of such docked complexes do not resemble the pose of aripiprazole with the crucial salt bridge to D116, but are stabilized by completely different intermolecular interactions.

Based on the local search results for the 5-HT_1A_ receptor combined with the experimental results presented in the next section, we can identify critical structural features in this family of ligands responsible for the high affinity of compounds **1**–**10** and low affinity of compounds **11**–**14** (see Figure 4). Clearly, the 6-acetyl-7-hydroxy-4-methylcoumarin scaffold fits well in the part of the 5-HT_1A_ receptor binding site located close to transmembrane helices 2 and 7 (as in the case of aripiprazole), but making an additional strong hydrogen bond between the acetyl moiety and Q97. The middle part of the ligand anchors the ligand in the binding site due to a strong salt bridge to D116. The varied affinity results come from the variable part of the phenylpiperazine moiety. Here, small substituents in all possible positions (ortho, meta and para) of the phenyl ring likely do not form any additional interactions, as this part of the binding site (helices 3, 5 and 6) is mostly hydrophobic. On the other hand the presence of two substituents (**10**–**11**) lowers the affinity, likely due to the too large size of the entire ligand, causing steric hindrances inside the binding pocket. A similar cause of lowering the affinity to the 5-HT_1A_ receptor can be attributed to compound **12** with a larger –NO_2_ substituent in the para position; on the other hand, compound **13** is likely to bind weakly due to the missing favorable interactions between the missing phenyl moiety and the hydrophobic pocket of helices 3, 5 and 6. In the case of compounds **12**–**14**, the low affinity may also be an effect of electron-withdrawing properties of substituents, which diminish the basic character of the anchoring nitrogen atom of the piperazine.

Finally, we performed analogous calculations to estimate affinities for 5-HT_2A_ receptor affinity; see Table 4. Here, the computational results are in disagreement with the experimental data, as molecular docking predicts very high affinities, but the experimental values are rather low. The computational estimate of Ki of aripiprazole (0.85 nM) is very close to the experimental value of 3.4–3.5 nM [21,22,23]; thus, one could expect similar values for the studied compounds due to the structural similarities to aripiprazole. This is clearly not the case, and for now, we do not have a good explanation for this discrepancy between computational estimates and experimental values, although the most likely cause is the overestimation of the non-covalent interactions by our docking approach. This is a common problem in all docking protocols, as most available software is rather accurate in predicting correct ligand poses but fail in accurate estimation of the binding affinities [24,25].

### 2.4. Biological Evaluation

All arylpiperazinyl derivatives of 6-acetyl-7-hydroxy-4-methylchromen-2-one described in this study (**1**–**15**) were tested for their affinity for the 5-HT_1A_ and 5-HT_2A_ receptors, as shown in Table 4. The results show that some of the synthesized systems have affinities in the nanomolar range toward 5-HT_1A_ and the low micromolar range toward 5-HT_2A_ receptors. 

As in our previous studies, we clearly see the influence of different substituents on the affinities of coumarin derivatives [9,10]. Compounds **3**–**7** with (2-bromophenyl)piperazinyl, (3-bromophenyl)piperazinyl, (4-bromophenyl)piperazinyl, (2-fluorophenyl)piperazinyl and (2-chlorophenyl)piperazinyl moieties showed high affinities for the 5-HT_1A_ receptors. Among these derivatives, compounds 6-acetyl-7-{4-[4-(3-bromophenyl)piperazin-1-yl]butoxy}-4-methylchromen-2-one (**4**) and 6-acetyl-7-{4-[4-(2-chlorophenyl)piperazin-1-yl]butoxy}-4-methylchromen-2-one (**7**) displayed the highest affinities for the 5-HT_1A_ receptor, with Ki values of 0.78 (0.4–1.4) nM and 0.57 (0.2–1.3) nM, respectively, nearby to the Ki values of 8-OH-DPAT (0.25 (0.097–0.66) nM). The introduction of the chloro substituents in the *ortho* position, or bromo substituents in the *meta* position of the phenyl ring increases affinities for the 5-HT_1A_ receptor. The replacement of the chlorine group in the 2-postion of the phenyl ring with the bromine or fluorine moieties displayed a decrease in affinities for the 5-HT_1A_ receptor form Ki = 0.57 nM to Ki = 1.96 nM or 1.04 nM for compounds **7**, **3** and **6**, respectively. Replacing the 2-position halogen with a methoxy group results in similar affinities. The situation is analogous when the bromine in position 3 of the phenyl ring of piperazine is replaced with a methoxy group, lowering the affinity from Ki = 0.78 nM for compound **4** to Ki = 12.9 nM for compound **2**. This relation is interesting due to the fact that for derivatives containing an acetyl group in the 8-position of the coumarin ring and a methoxy group in the 2- or 3-position of the piperazine phenyl ring, we have previously demonstrated excellent affinity for the 5-HT_1A_ receptor [9,10]. Changing the position of the acetyl group from 8 to 6 in the coumarin ring lowers the affinity of the derivatives with the methoxy substituent from Ki = 1.0 nM to Ki = 5.75 nM for the 2-position of the methoxy moiety and from Ki = 0.8 nM to Ki = 12.9 for the 3-position of methoxy moiety. The affinity for the 5-HT_1A_ receptor for derivatives containing bromo, fluoro or chloro substituents in the *ortho* position of the phenyl ring of the piperazine is analogous for both 6-acetyl and 8-acetyl coumarin. In the case of (3-bromophenyl)piperazine, 5-HT_1A_ receptor affinity increases by threefold for 6-acetylcoumarin compared to 8-acetylcoumarin (Ki = 2.5 nM to Ki = 0.78 nM). In the case of (2-fluorophenyl)piperazine, the affinity remains similar regardless of the position of the acetyl group in the coumarin ring (Ki = 1 nM for 8-acetylcoumarin and Ki = 1.04 nM for 6-acetylcoumarin). Surprisingly, we also obtained high affinity for compound **5** with bromine in the *para* position of the piperazine phenyl ring (Ki = 1.40 nM). An introduction of the nitro-substituents in the *para* position or the replacement of the phenyl ring with a heterocyclic moiety produced a decrease in 5-HT_1A_ receptor affinity (Ki = 1658–9617 nM), in line with the trend described in our previous works. 

For the 5-HT_1A_ receptor, we also determined the agonist or antagonist properties of all of the new compounds, as shown in Table 5. Our results confirmed the 5-HT_1A_ antagonistic properties for derivatives **1**, **3**, **5**, and **11**–**14**. The strongest antagonist in this group was 6-acetyl-7-{4-[4-(2-methoxyphenyl)piperazin-1-yl]butoxy}-4-methylchromen-2-one (**1**) with an IC_50_ value of 301 nM, which is twenty times higher than for WAY 100636, the antagonist of the 5-HT_1A_ receptor used as a reference substance. Curiously, the 6-acetyl-7-{4-[4-(3-methoxyphenyl)piperazin-1-yl] butoxy}-4-methylchromen-2-one (**2**) derivative, which differs only in the position of the methoxy group in the phenyl ring of piperazine, showed agonistic activity at the 5-HT_1A_ receptor, although it was almost five times weaker than 8-OH-DPAT used as the reference compound (EC_50_ = 181 nM for 2 and EC_50_ = 38 for 8-OH-DPAT). The agonistic profile was also observed for compounds **4**, **6**–**10** and **15**. 6-Acetyl-7-{4-[4-(2-chlorophenyl)piperazin-1-yl]butoxy}-4-methylchromen-2-one (**7**), which showed the highest affinity for the 5-HT_1A_ receptor, also turned out to be the strongest agonist among the group of compounds tested with an EC_50_ = 49 nM. Furthermore, we verified the relationship between the Ki and IC_50_/EC_50_ values for compounds **1**–**15** and found that there is a positive moderate correlation for both quantities (correlation coefficients R^2^ = 0.58 for Ki/IC_50_ and R^2^ = 0.66 for Ki/EC_50_, respectively). This result shows that Ki is a good preliminary indicator of the biological activity of these compounds, but functional tests are necessary to verify the functional nature of this activity.

All compounds studied showed low, micromolar affinities for the 5-HT_2A_ receptor. Compounds bearing the (3,5-dimethylphenyl) and (2-cyanophenyl)piperazinyl piperazinyl moieties (**10** and **8**, respectively) showed the highest affinities in the whole series (Ki values of 516 nM or 705 nM, respectively). However, their affinities for the 5-HT_2A_ receptor were more than two orders of magnitude weaker than ketanserin, which served as the reference compound (Ki value of 1.33 nM). In both cases of **10** and **8**, the position of the acetyl group in the coumarin ring had a significant influence on the action at the 5-HT_2A_ receptor, as the Ki values of analogous compounds with the acetyl group in position 8 of the coumarin ring were respectively found to be much lower at 58 and 91 nM, respectively [10,11]. 

## 3. Materials and Methods

### 3.1. Chemistry

All starting materials were purchased from Aldrich or Merck and were used without further purification. The Plazmatronika 1000 microwave oven was used (http://www.plazmatronika.com.pl (accessed on 27 December 2020). The melting points were determined with ElectroThermal 9001 Digital Melting Point apparatus and are uncorrected. High resolution mass spectra were recorded on a Micromsass LCT (ESI-TOF). ^1^H NMR, ^13^C NMR spectra in solution were recorded at 25 °C with a Varian NMRS-300 spectrometer, and standard Varian VnmrJ 2.1B software was employed. The calculated shielding constants were used as an aid in assigning resonances of 13C atoms. Chemical shifts δ (ppm) were referenced to TMS. TLC was carried out using Kieselgel 60 F254 sheets, eluent: CHCl_3_: MeOH; 10:0.25 and spots were visualized by UV e 254 and 365 nm.

Compounds **A** and **1–15** were prepared in accordance with the previously reported procedures [9]. Atom numbering, ^1^H NMR and ^13^C NMR spectra of all synthesized compounds are available in the ESI. 


*6-Acetyl-7-(4-bromobutoxy)-4-methylchromen-2-one (*
**A**
*)*


M.p.: 114 °C, Rf = 0.32, yield 81%, ^1^H NMR (300 MHz, CHCl_3_) δ ppm: 8.07 (s, 1H, H-5), 6.86 (s, 1H, H-8), 6.20 (s, 1H, H-3), 4.19 (t, *J* = 6.0 Hz, 2H, H-1′), 3.52 (t, *J* = 6.0 Hz, 2H, H-4′), 2.67 (s, 3H, H-11), 2.45 (s, 3H, H-9), 2.12 (q, 4H, H-2′, H-3′); ^13^C NMR (75 MHz, CHCl_3_) δ ppm: 197.7 (C-10), 161.0 (C-7), 160.5 (C-2), 157.7 (C-8a), 152.9 (C-4), 128.1 (C-6), 124.9 (C-5), 113.6 (C-3), 112.9 (C-4a), 100.5 (C-8), 68.6 (C-1′), 33.1 (C-4′), 32.3 (C-3′), 29.5 (C-2′), 27.8 (C-11), 18.9 (C-9); TOF MS ES+: [M + Na]^+^ calcd for C_16_H_17_O_4_NaBr: 375.0208 found 375.0196.


*6-Acetyl-7-{4-[4-(2-methoxyphenyl)piperazin-1-yl]butoxy}-4-methylchromen-2-one (*
**1**
*)*


M.p.: 126–127 °C, Rf = 0.38, yield 95%, ^1^H NMR (300 MHz, CDCl_3_) δ ppm: 8.08 (s, 1H, H-5), 6.98 (m, 5H, H-8, H-3”-H-6”), 6.19 (s, 1H, H-3), 4.19 (t, *J* = 6.0 Hz, 2H, H-1′), 3.86 (s, 3H, H-7”), 3.13 (br. s, 4H, H-3p, H-5p), 2.69 (br. s, 7H, H-11, H-2p, H-6p), 2.54 (t, *J* = 6.0 Hz, 2H, H-4′), 2.45 (s, 3H, H-9), 2.00 (m, 2H, H-3′), 1,81 (m, 2H, H-2′); ^13^C NMR (75 MHz, CDCL_3_) δ ppm: 197.9 (C-10), 161.3 (C-6”), 160.6 (C-7), 157.7 (C-2), 153.0 (C-8a), 152.4 (C-4), 141.3 (C-2”), 128.2 (C-6), 125.0 (C-5), 123.2 (C-6”), 121.2 (C-4”), 118.4 (C-5”), 113.5 (C-3”), 112.9 (C-4a), 111.3 (C-3), 100.6 (C-8), 69.5 (C-1′), 58.3 (C-3p, C-5p), 55.6 (C-4′), 53.6 (C-7”), 50.8 (C-2p, C-6p), 32.4 (C-2′), 29.9 (C-11), 23.6 (C-3′), 18.9 (C-9); TOF MS ES+: [M + H]^+^ calcd for C_27_H_33_O_5_N_2_: 465.2389 found 465.2401


*6-Acetyl-7-{4-[4-(3-methoxyphenyl)piperazin-1-yl]butoxy}-4-methylchromen-2-one (*
**2**
*)*


M.p.: 83–85 °C, Rf = 0.27, yield 34%, ^1^H NMR (300 MHz, CDCl_3_) δ ppm: 8.07 (s, 1H, H-5), 7.20 (t, *J* = 7.5 Hz, 1H, H-5”), 6.87 (s, 1H, H-8), 6.56 (d, *J* = 9.0 Hz, 1H, H-4”), 6.47 (m, 2H, H-2”, H-6”), 6.20 (s, 1H, H-3), 4.19 (t, *J* = 6.0 Hz, 2H, H-1′), 3.81 (s, 3H, H-7”), 3.31 (br., s, 4H, H-3p, H-5p), 2.74 (br., s, 4H, H-2p, H-6p), 2.68 (s, 3H, H-11), 2.62 (br., s, 2H, H-4”), 2.45 (s, 3H, H-9), 2.01 (m, 2H, H-2′), 1.87 (br., s, 2H, H-3′); ^13^C NMR (75 MHz, CDCl_3_) δ ppm: 197.8 (C-10), 161.1 (C-7), 160.8 (C-2, C-3”), 160.6 (C-8a), 157.7 (C-4), 152.9 (C-1”), 130.2 (C-5”), 128.1 (C-6), 125.1 (C-5), 113.6 (C-4a), 113.1 (C-3), 109.4 (C-4”), 105.4 (C-6”), 103.1 (C-8), 100.6 (C-2”), 69.2 (C-1′), 57.9 (C-3p, C-5p), 55.3 (C-2p, C-6p), 52.9 (C-4′), 48.5 (C-2′), 48.3 (C-7”), 32.3 (C-3′), 26.9 (C-11), 18.9 (C-9); TOF MS ES+: [M + H]^+^ calcd for C_27_H_33_O_5_N_2_: 465.2389 found 465.2381.


*6-Acetyl-7-{4-[4-(2-bromophenyl)piperazin-1-yl]butoxy}-4-methylchromen-2-one (*
**3**
*)*


M.p.: 121–122 °C, Rf = 0.15, yield 79%, ^1^H NMR (300 MHz, CDCl_3_) δ ppm: 8.08 (s, 1H, H-5), 7.57 (dd, *J*_1_ = 12.0 Hz, *J_2_* = 3.0 Hz, 1H, H-6”), 7.33 (s, 1H, H-8), 7.12 (d, *J* = 9.0 Hz, 1H, H-3”), 6.96 (br.t, *J* = 9.0 Hz, 1H, H-4”), 6.88 (s, 1H, H-5”), 6.21 (s, 1H, H-3), 4.21 (t, *J* = 4.5 Hz, 2H, H-1′), 3.20 (br. s, 4H, H-3p, H-5p), 2.69 (s, 3H, H-11), 2.46 (s, 3H, H-9), 2.03 (br. s, 2H, H-4′), 1.59 (br.s, 6H, H-2p, H-6p, H-2′), 1.27 (br.s., 2H, H-3′); ^13^C NMR (75 MHz, CDCl_3_) δ ppm: 197.7 (C-10), 160.8 (C-7, C-2), 160.5 (C-4), 157.7 (C-8a), 152.9 (C-1”), 134.0 (C-3”), 128.8 (C-6, C-5), 128.1 (C-4), 125.1 (C-5”), 121.6 (C-6”), 120.0 (C-2”), 113.7 (C-3), 113.1 (C-4a), 100.6 (C-8), 68.8 (C-1′), 57.6 (C-4′), 53.0 (C-3p, C-5p), 52.9 (C-2p, C-6p), 32.3 (C-2′), 29.9 (C-3′), 26.8 (C-11), 18.9 (C-9); TOF MS ES+: [M + Na]^+^ calcd for C_26_H_29_BrO_4_N_2_Na: 535.1208 found 535.1195.


*6-Acetyl-7-{4-[4-(3-bromophenyl)piperazin-1-yl]butoxy}-4-methylchromen-2-one (*
**4**
*)*


M.p.: 109–110 °C, Rf = 0.16, yield 82%, ^1^H NMR (300 MHz, CDCl_3_) δ ppm: 8.05 (s, 1H, H-5), 7.10 (t, *J* = 9.0 Hz, 1H, H-5”), 7.03 (m, 1H, H-6”), 6.96 (m, 1H, H-4”), 6.81 (m, 2H, H-8, H-2”), 6.17 (s, 1H, H-3), 4.17 (t, *J* = 6.0 Hz, 2H, H-1′), 3.23 (t, *J* = 4.5 Hz, 4H, H-3p, H-5p), 2.64 (m, 7H, H-11, H-2p, H-6p), 2.53 (t, *J* = 7.5 Hz, 2H, H-4′), 2.42 (s, 3H, H-9), 1.99 (m, 2H, H-2′), 1.78 (m, 2H, H-3′); ^13^C NMR (75 MHz, CDCl_3_) δ ppm: 197.8 (C-10), 161.2 (C-7), 160.5 (C-2), 157.7 (C-4), 152.9 (C-8a), 152.3 (C-1”), 130.5 (C-5”), 128.1 (C-6), 125.0 (C-5), 123.4 (C-3”), 122.6 (C-4”), 118.9 (C-2”, C-6”), 114.6 (C-4a), 113.5 (C-3), 100.5 (C-8), 69.3 (C-1′), 57.9 (C-4′), 53.0 (C-3p, C-5p), 48.5 (C-2p, C-6p), 32.3 (C-2′), 27.0 (C-3′), 23.3 (C-11), 18.9 (C-9); TOF MS ES+: [M + Na]^+^ calcd for C_26_H_29_BrO_4_N_2_Na: 535.1208 found 535.1197.


*6-Acetyl-7-{4-[4-(4-bromophenyl)piperazin-1-yl]butoxy}-4-methylchromen-2-one (*
**5**
*)*


M.p.: 145–146 °C, Rf = 0.26, yield 44%, ^1^H NMR (300 MHz, CDCl_3_) δ ppm: 8.07 (s, 1H, H-5), 7.35 (d, *J =* 9.0 Hz, 2H, H-3”, H-5”), 6.87 (s, 1H, H-8), 6.82 (d, *J =* 9.0 Hz, 2H, H-2”, H-6”), 6.20 (s, 1H, H-3), 4.19 (t, *J =* 6.0 Hz, 2H, H-1′), 3.25 (br. s, 4H, H-3p, H-5p), 6.28 (br. s, 6H, H-2p, H-6p, H-4′), 2.45 (s, 3H, H-9), 2.02 (m, 2H, H-2′), 1.96 (m, 2H, H-3′); ^13^C NMR (75 MHz, CDCL_3_) δ ppm: 197,8 (C-10), 161,1 (C-7, C-2), 160.5 (C-8a), 157.7 (C-4), 152.9 (C-1”), 132.1 (C-3”, C-5”), 128.1 (C-6), 125.0 (C-5), 118.0 (C-4”), 113.5 (C-6”, C-2”), 113.0 (C-4a, C-3), 100.6 (C-8), 69.3 (C-1′), 58.0 (C-3p, C-5p), 53.0 (C-4′), 48.7 (C-2p, C-6p), 32.3 (C-2′, C-3′), 27.0 (C-11), 18.9 (C-9); TOF MS ES+: [M + H]^+^ calcd for C_26_H_30_BrO_4_N_2_: 513.1389 found 513.1398.


*6-Acetyl-7-{4-[4-(2-fluorophenyl)piperazin-1-yl]butoxy}-4-methylchromen-2-one (*
**6**
*)*


M.p.: 112–113 °C, Rf = 0.32, yield 95%, ^1^H NMR (300 MHz, CDCl_3_) δ ppm: 8.07 (s, 1H, H-5), 7.01 (m, 4H, H-3”, H-4”, H-5”, H-6”), 6.87 (s, 1H, H-8), 6.20 (s, 1H, H-3), 4.20 (t, *J* = 6.0 Hz, 2H, H-1′), 3.23 (br. s, 4H, H-3p, H-5p), 2.81 (br. s, 4H, H-2p, H-6p), 2.68 (s, 3H, H-11), 2.64 (br. s. 2H, H-4′), 2.45 (s, 3H, H-9), 2.03 (m, 2H, H-2′), 1.90 (m, 2H, H-3′); ^13^C NMR (75 MHz, CDCL_3_) δ ppm: 197.8 (C-10), 161.0 (C-3”), 160.6 (C-7), 157.7 (C-2), 157.5 (C-8a), 154.2 (C-4), 152.9 (C-1”), 128.1 (C-5”), 125.1 (C-6). 124.8 (C-5), 123.5 (C-4a), 119.4 (C-3), 116.5 (C-4”), 116.3 (C-6”), 113.6, 113.1 (C-8), 100.6 (C-2”), 69.1 (C-1′), 57.8 (C-4”), 53.0 (C-2p, C-6p), 49.2 (C-3p, C-5p), 32.3 (C-2′), 29.9 (C-3′), 26.9 (C-11), 18.9 (C-9); TOF MS ES+: [M + H]^+^ calcd for C_26_H_30_FO_4_N_2_: 453.2190 found 453.2198.


*6-Acetyl-7-{4-[4-(2-chlorophenyl)piperazin-1-yl]butoxy}-4-methylchromen-2-one (*
**7**
*)*


M.p.: 132–133 °C, Rf = 0.19, yield 64%,^1^H NMR (300 MHz, CDCl_3_) δ ppm: 8.07 (s, 1H, H-5), 7.37 (d, *J* = 9.0 Hz, 1H, H-6”), 7.27 (d, *J* = 9.0 Hz, 1H, H-3”), 7.03 (m, 2H, H-4”, H-5”),6.87 (s, 1H, H-8), 6.19 (s, 1H, H-3), 4.20 (t, *J* = 6.0 Hz, 2H, H-1′), 3.18 (br. s, 4H, H-3p, H-5p), 2.78 (br. s, 4H, H-2p, H-6p), 2.69 (s, 3H, H-11), 2.64 (br. s, 2H, H-4′), 2.01 (m, 2H, H-2′), 1.86 (m, 2H, H-3′); ^13^C NMR (75 MHz, CDCl_3_) δ ppm: 197.8 (C-10), 161.1 (C-7), 160.6 (C-2), 157.7 (C-8a), 152.9 (C-5, C-1”), 130.8 (C-3”), 128.9 (C-2”), 128.2 (C-5”), 127.9 (C-5), 125.1 (C-6), 124.3 (C-4”), 120.7 (C-6”), 113.6 (C-3), 113.0 (C-4a), 100.6 (C-8), 69.2 (C-1′), 58.0 (C-3p, C-5p), 53.4 (C-2p, C-6p), 50.5 (C-4′), 32.3 (C-2′), 27.0 (C-3′), 22.9 (C-11), 18.9 (C-9); TOF MS ES+: [M + H]^+^ calcd for C_26_H_30_ ClO_4_N_2_: 469.1894 found 469.1898.


*6-Acetyl-7-{4-[4-(2-cyanophenyl)piperazin-1-yl]butoxy}-4-methylchromen-2-one (*
**8**
*)*


M.p.: 117–118 °C, Rf = 0.22, yield 31%, ^1^H NMR (300 MHz, CDCl_3_) δ ppm: 8.05 (s, 1H, H-5), 7.57 (m, 2H, H-3”, H-5”), 7.10 (m, 2H, H-6”, H-4”), 6.85 (s, 1H, H-8), 6.20 (s, 1H, H-3), 4.20 (t, *J* = 4.5 Hz, 2H, H-1′), 3.47 (br. s, 10H, H-2p, H-3p, H-5p, H-6p, H-4′), 2.67 (s, 3H, H-11), 2.44 (s, 3H, H-9), 2.04 (br., s, 2H, H-2′), 1.61 (br., s, 2H, H-3′); ^13^C NMR (75 MHz, CDCl_3_) δ ppm: 197.7 (C-10), 160.5 (C-2, C-7), 157.7 (C-4, C-8a), 152.9 (C-1”), 134.4 (C-3”, C-4”), 128.1 (C-5, C-6), 125.1 (C-4”, C-6”, C-7”), 113.7 (C-3), 113.2 (C-4a), 100.6 (C-2”, C-8), 68.8 (C-1′), 57.5 (C-4′, C-2p, C-6p), 52.8 (C-3p, C-5p), 32.2 (C-2′), 29.9 (C-3′), 26.7 (C-9), 18.9 (C-11); TOF MS ES+: [M + H]^+^ calcd for C_27_H_30_O_4_N_3_: 460.2236 found 460.2233.


*6-Acetyl-7-{4-[4-(2,3-dichlorophenyl)piperazin-1-yl]butoxy}-4-methylchromen-2-one (*
**9**
*)*


M.p.: 111–112 °C, Rf = 0.37, yield 57%,^1^H NMR (300 MHz, CDCl_3_) δ ppm: 8.07 (s, 1H, H-5), 7.17 (m, 2H, H-6”, H-4”), 6.98 (m, 1H, H-5”), 6.87 (s, 1H, H-8), 6.19 (s, 1H, H-3), 4.20 (t, *J* = 7.5 Hz, 2H, H-1′), 3.13 (br. s, 4H, H-3p, H-5p), 2.72 (br.s, 4H, H-2p, H-6p), 2.68 (s, 3H, H-11), 2.58 (m, 2H, H-4′), 2.45 (s, 3H, H-9), 2.00 (m, 2H, H-2′), 1.83 (m, 2H, H-3′); ^13^C NMR (75 MHz, CDCl_3_) δ ppm: 197.8 (C-10), 161.2 (C-2, C-7), 160.6 (C-8a), 157.7 (C-4), 152.9 (C-2”), 134.2 (C-3”), 128.1 (C-5”), 127.8 (C-1”), 125.2 (C-5), 125.0 (C-6), 118.9 (C-4”), 113.5 (C-6”), 112.9 (C-3, C-4a), 100.6 (C-8), 69.2 (C-1′), 57.9 (C-4′), 53.3 (C-3p, C-5p), 50.7 (C-2p, C-6p), 32.3 (C-2′), 26.9 (C-3′), 23.0 (C-11), 18.9 (C-9); TOF MS ES+: [M + Na]^+^ calcd for C_26_H_28_Cl_2_O_4_N_2_Na: 525.1324 found 525.1313.


*6-Acetyl-7-{4-[4-(3,5-dimethylphenyl)piperazin-1-yl]butoxy}-4-methylchromen-2-one (*
**10**
*)*


M.p.: 101–102 °C, Rf = 0.20, yield 67%,^1^H NMR (300 MHz, CDCl_3_) δ ppm: 8.07 (s, 1H, H-5), 6.86 (s, 1H, H-5), 6.58 (m, 3H, H-2”, H-4”, H-6”), 6.19 (s, 1H, H-3), 4.19 (t, *J* = 6.0 Hz, 2H, H-1′), 3.26 (br. s, 4H, H-3p, H-5p), 2.71 (br.s, 4H, H-2p, H-6p), 2.68 (br. s, 3H, H-11), 2.58 (m, 2H, H-4′), 2.45 (s, 3H, H-9), 2.29 (s, 6H, H-7”, H-8”), 2.00 (m, 2H, H-2′), 1.86 (m, 2H, H-3′); ^13^C NMR (75 MHz, CDCl_3_) δ ppm: 197.7 (C-10), 161.2 (C-7), 160.5 (C-2), 157.7 (C-8a), 152.9 (C-4), 151.3 (C-1”), 138.7 (C-3”, C-5”), 128.1 (C-5), 124.9 (C-6), 121.9 (C-4”), 114.2 (C-2”, C-6”), 113.3 (C-4a), 112.8 (C-3), 100.5 (C-8), 69.3 (C-1′), 58.1 (C-4′), 53.4 (C-3p, C-5p), 49.2 (C-2p, C-6p), 32.3 (C-2′), 29.8 (C-3′), 27.1 (C-11), 23.4 (C-7”), 21.8 (C-8”), 18.8 (C-9); TOF MS ES+: [M + H]^+^ calcd for C_28_H_35_O_4_N_2_: 436.2597 found 436.2604.


*6-Acetyl-7-{4-[4-(2,5-dimethylphenyl)piperazin-1-yl]butoxy}-4-methylchromen-2-one (*
**11**
*)*


M.p.: 130–131 °C, Rf = 0.28, yield 49%, ^1^H NMR (300 MHz, CDCl_3_) δ ppm: 8.07 (s, 1H, H-5), 7.08 (d, *J* = 6.0 Hz, 1H, H-3”), 6.87 (s, 1H, H-8), 6.83 (d, *J*= 9.0 Hz, 2H, H-4”, H-6”), 6.20 (s, 1H, H-3), 4.20 (t, *J*= 6.0 Hz, 2H, H-1′), 3.07 (br. s, 4H, H-3p, H-5p), 2.69 (s, 3H, H-11), 2.45 (s, 3H, H-9), 2.32 (s, 3H, H-7”), 2.26 (s, 3H, H-8”), 2.02 (m, 2H, H-2′), 1.92 (m, 2H, H-3′); ^13^C NMR (75 MHz, CDCL_3_) δ ppm: 197.8 (C-10), 160.9 (C-7), 160.5 (C-2), 157.7 (C-8a), 152.9 (C-4), 136.6 (C-1”), 131.1 (C-5”), 129.4 (C-2”), 128.1 (C-3”), 125.1 (C-5), 124.8 (C-6), 120.4 (C-4”), 113.6 (C-6”), 113.1 (C-3, C-4a), 100.6 (C-8), 69.0 (C-1′), 57.8 (C-4′), 53.5 (C-3p, C-5p), 50.4 (C-2p, C-6p), 32.3 (C-2′), 29.9 (C-11), 26.9 (C-3′), 22.4 (C-7”), 18.9 (C-9), 17.5 (C-8”); TOF MS ES+: [M + H]^+^ calcd for C_28_H_35_O_4_N_2_: 463.2597 found 463.2599.


*6-Acetyl-7-{4-[4-(4-nitrophenyl)piperazin-1-yl]butoxy}-4-methylchromen-2-one (*
**12**
*)*


M.p.: 138–139 °C, Rf = 0.44, yield 47%, ^1^H NMR (300 MHz, CDCl_3_) δ ppm: 8.14 (d, *J* = 9.0 Hz, 2H, H-3”, H-5”), 8.07 (s, 1H, H-5), 6.87 (s, 1H, H-8), 6.84 (d, *J*= 9.0 Hz, 2H, H-2”, H-6”), 6.20 (s, 1H, H-3), 4.20 (t, *J* = 6.0 Hz, 2H, H-1′), 3.47 (br. s, 4H, H-3p, H-5p), 2.68 (s, 3H, H-11), 2.62 (m, 6H, H-4′,H-2p, H-6p), 2.01 (m, 2H, H-2′), 1.79 (m, 2H, H-3′); ^13^C NMR (75 MHz, CDCL_3_) δ ppm: 197.9 (C-10), 160.6 (C-7, C-2), 157.7 (C-8a), 152.9 (C-1”), 152.9 (C-4), 128.1 (C-4”), 126.2 (C-5, C-6), 125.1 (C-3”, C-5”), 113.6 (C-3, C-4a), 113.1 (C-2”, C-6”), 100.6 (C-8), 69.2 (C-1′), 57.9 (C-4′), 52.7 (C-3p, C-5p), 46.9 (C-2p, C-6p), 32.4 (C-2′), 32.3 (C-3′), 26.9 (C-11), 18.9 (C-9); TOF MS ES+: [M + Na]^+^ calcd for C_26_H_29_O_6_N_3_Na: 502.1954 found 502.1942.


*6-Acetyl-7-[4-(morpholin-4-yl)butoxy]-4-methylchromen-2-one (*
**13**
*)*


M.p.: 115–116 °C, Rf = 0.29, yield 69%, ^1^H NMR (300 MHz, CDCl_3_) δ ppm: 8.07 (s, 1H, H-5), 6.86 (s, 1H, H-8), 6.19 (s, 1H, H-8), 4.17 (t, *J* = 7.5 Hz, 2H, H-1′), 3.79 (br. s, 4H, H-3m, H-5m), 2.67 (s, 3H, H-11), 2.54 (br. s, 6H, H-4′, H-2m, H-6m), 2.45 (s, 3H, H-9), 1.98 (m, 2H, H-2′), 1.79 (m, 2H, H-3′); ^13^C NMR (75 MHz, CDCL_3_) δ ppm:197.8 (C-10), 161.2 (C-7), 160.6 (C-2), 157.7 (C-8a), 152.9 (C-4), 128.1 (C-6), 125.1 (C-5), 113.6 (C-4a), 113.0 (C-3), 100.6 (C-8), 69.3 (C-1′), 66.5 (C-3m, C-5m), 58.4 (C-2m, C-6m), 53.6 (C-4′), 32.3 (C-2′), 26.9 (C-3′), 22.8 (C-11), 18.9 (C-9); TOF MS ES+: [M + Na]^+^ calcd for C_20_H_25_NO_5_Na: 382.1630 found 382.1618.


*6-Acetyl-7-{4-(4-piridin)piperazin-1-yl}butoxy}-4-methylchromen-2-one (*
**14**
*)*


M.p.: 107–108 °C, Rf = 0.19, yield 63%, ^1^H NMR (300 MHz, CDCl_3_) δ ppm: 8.21 (br. s, 2H, H-3”, H-5”), 8.07 (s, 1H, H-5), 6.88 (s, 1H, H-8), 6.84 (br. d, *J* = 9.0 Hz, 2H, H-2”, H-6”), 6.20 (s, 1H, H-3), 4.20 (t, *J* = 6.0 Hz, 2H, H-1′), 3.59 (t, *J*= 4,5 Hz, 4H, H-3p, H-5p), 2.68 (s, 3H, H-11), 2.64 (t, *J* = 6 Hz, 4H, H-2p, H-6p), 2.53 (t, *J* = 7,5 Hz, 2H, H-4′), 2.46 (s, 1H, H-9), 2.00 (m, 2H, H-2′), 1.77 (m, 2H, H-3′); ^13^C NMR (75 MHz, CDCL_3_) δ ppm:197.9 (C-10), 161.2 (C-7), 160.6 (C-2), 157.8 (C-8a), 156.5 (C-4), 153.0 (C-1”), 143.1 (C-3”, C-5”), 128.2 (C-6), 125.1 (C-5), 113.6 (C-3), 113.1 (C-4a), 107.6 (C-2”, C-6”), 100.6 (C-8), 69.3 (C-1′), 57.8 (C-3p, C-5p), 52.5 (C-2p, C-6p), 46.3 (C-4′), 32.3 (C-2′), 29.3 (C-3′), 23.5 (C-11), 18.9 (C-9); TOF MS ES+: [M + H]^+^ calcd for C_25_H_30_O_4_N_3_: 436.2236 found 435.2247.


*6-Acetyl-7-{4-[4-(pyrazin-2-yl)piperazin-1-yl]butoxy}-4-methylchromen-2-one (*
**15**
*)*


M.p.: 150–151 °C, Rf = 0.35, yield 69%,^1^H NMR (300 MHz, CDCl_3_) δ ppm: 8.16 (s, 1H, H-6”), 8.08 (s, 2H, H-3”, H-4”), 7.87 (s, 1H, H-5), 6.86 (s, 1H, H-8), 6.18 (s, 1H, H-3), 4.19 (t, *J* = 6.0 Hz, 2H, H-1′), 3.66 (br. s, 4H, H-3p, H-5p), 2.67 (s, 3H, H-11), 2.63 (br. s, 4H, H-2p, H-6p), 2.55 (br., s, 2H, H-4′), 2.00 (m, 2H, H-2′), 1.81 (m, 2H, H-3′); ^13^C NMR (75 MHz, CDCl_3_) δ ppm: 197.8 (C-10), 161.3 (C-7), 160.6 (C-2), 157.4 (C-8a), 155.1 (C-1”), 153.0 (C-4), 141.9 (C-3”), 133.2 (C-4”), 131.2 (C-6”), 128.2 (C-6), 125.0 (C-5), 113.5 (C-4a), 112.9 (C-3), 100.5 (C-8), 69.4 (C-1′), 58.2 (C-4′), 52.9 (C-3p, C-5p), 44.6 (C-2p, C-6p), 32.3 (C-2′), 27.0 (C-3′), 23.5 (C-11), 18.9 (C-9); TOF MS ES+: [M + Na]^+^ calcd for C_24_H_28_O_4_N_4_Na: 459.2008 found 459.2018.

### 3.2. X-ray Crystallography

The X-ray measurement of **2** was performed at 130.0(5) K on a Bruker D8 Venture PhotonII diffractometer equipped with a TRIUMPH monochromator and a MoKα fine focus sealed tube (*λ* = 0.71073 Å). A total of 2690 frames were collected with the Bruker APEX3 program [26]. The frames were integrated with the Bruker SAINT, V8.40A software package [27] using a narrow-frame algorithm. Integration of the data using a monoclinic unit cell yielded a total of 62369 reflections to a maximum *θ* angle of 28.50° (0.74 Å resolution), of which 5948 were independent (average redundancy 10.486, completeness = 99.9%, *R_int_* = 2.78%, *R_sig_* = 1.41%) and 5240 (88.10%) were greater than 2*σ*(*F*^2^). The final cell constants of *a* = 24.3110(12) Å, *b* = 12.0631(6) Å, *c* = 8.0498(4) Å, *β* = 96.000(2)°, *V* = 2347.8(2) Å^3^ are based upon the refinement of the XYZ-centroids of 5239 reflections above 20 *σ*(*I*) with 4.042° < 2*θ* < 60.45°. Data were corrected for absorption effects using the Multi-Scan method (SADABS) [28]. The ratio of minimum to maximum apparent transmission was 0.943. The calculated minimum and maximum transmission coefficients (based on crystal size) are 0.947 and 0.991.

The structure was solved and refined using the SHELXTL Software Package [29,30] using the space group *P*2_1_/*c*, with *Z* = 4 for the formula unit, C_27_H_32_N_2_O_5_. The final anisotropic full-matrix least-squares refinement on *F*^2^ with 311 variables converged at *R*1 = 3.68% for the observed data and *wR*2 = 10.58% for all data. The goodness of fit was 1.031. The largest peak in the final difference electron density synthesis was 0.355 e^−^/Å^3^, and the largest hole was −0.217 e^−^/Å^3^ with an RMS deviation of 0.042 e^−^/Å^3^. Based on the final model, the calculated density was 1.314 g/cm^3^ and *F*(000), 992 e^−^. The details concerning the crystal data and structural parameters of **2** are collected in Table 1.

All non-hydrogen atoms were refined anisotropically. All hydrogen atoms were placed in calculated positions and refined within the riding model. The temperature factors of the hydrogen atoms were not refined and were set at 1.2 (C_ar_-H, CH_2_ groups) or 1.5 (CH_3_ group) times higher than the *U_eq_* of the corresponding heavy atom. The atomic scattering factors were taken from the International Tables [31]. Molecular graphics was prepared using the program Mercury 2020.2.0 [32].

CCDC 2213451 contains the supplementary crystallographic data for this study. The data can be obtained free of charge from The Cambridge Crystallographic Data Centre via www.ccdc.cam.ac.uk/structures.

### 3.3. Biological Evaluation

#### 3.3.1. Membrane Preparation

Sprague–Dawley rats were sacrificed by isoflurane overdose. Brains were rapidly removed and placed on ice. Hippocampi (for 5-HT_1A_ assays) and frontal cortices (for 5-HT_2A_ assays) were dissected on a Petri dish. The tissue from 10 rats was homogenized in 30 vol. homogenization buffer (50 mM Tris-HCl, pH = 4.7, 1 mM EDTA, 1 mM dithiothreitol) with a hand-held Teflon-glass homogenizer. The homogenate was centrifuged at 48,000× *g* at 4 °C for 15 min. The pellet was suspended and homogenized in homogenization buffer and incubated for 10 min at 36 °C. The centrifugation and suspension steps were repeated twice. The final pellet was homogenized in 5 vol. 50 mM Tris-HCl, pH = 7.4 buffer and stored at −80 °C for not more than 6 months.

#### 3.3.2. Competitive 5-HT_1A_ and 5-HT_2A_ Binding Assays

For the 5-HT_1A_ assay, ten concentrations equally spaced on a logarithmic scale (10^−14^M−10^−5^M) of each compound were incubated in duplicate with 1 nM [^3^H]8-OH-DPAT (specific activity: 200 Ci/mmol, Perkin Elmer, Waltham, MA, USA) for 60 min at 36 °C in a 50 mM Tris-HCl buffer (pH 7.4), supplemented with 0.1% ascorbate, 5 mM MgCl_2_ and 80 µg of rat hippocampal membrane suspension. For the 5-HT_2A_ assay, 160 µg of rat frontal cortex membrane suspension was incubated with 1 nM [^3^H]ketanserin (specific activity: 22.8 Ci/mmol, Perkin Elmer, Waltham, MA, USA) for 60 min at 36 °C in a 50 mM Tris-HCl (pH 7.4) buffer, supplemented with 0.1% ascorbate and 3 mM CaCl_2_. Non-specific binding was determined with 10 μM serotonin in both assays. The final DMSO concentration in the assay was 5%. After incubation, the reaction mixture was deposited with the FilterMate-96 Harvester (Perkin Elmer, Waltham, MA, USA) onto Unifilter^®^ GF/C plates (Perkin Elmer, Waltham, MA, USA) presoaked in 0.4% PEI for 1h. Each well was washed with 2 mL of 50 mM Tris-HCl (pH 7.4) buffer to separate bound ligands from free ones. Plates were left to dry overnight. Then, 35 µL of Microscint-20 scintillation fluid (Perkin Elmer, Waltham, MA, USA) was added to each filter well and left to equilibrate for 2 h. Filter-bound radioactivity was counted in a MicroBeta^2^ LumiJet scintillation counter (Perkin Elmer, Waltham, MA, USA). Binding curves were fitted with one-site non-linear regression. Binding affinity (pKi ± SEM and Ki ± 95% confidence intervals) for each compound was calculated from EC_50_ values with the Cheng-Prusoff equation from two separate experiments.

#### 3.3.3. 5-HT_1A_ Receptor Activation in the [^35^S]GTP-γ-S Assay

Ten compound concentrations equally spaced on a log scale (10^−4.5^ M to 10^−9^ M) were incubated in duplicate with rat hippocampal membrane preparations (5 μg per well) in assay buffer (50 mM Tris-HCl, pH = 7.4, 1 mM EGTA, 3 mM MgCl_2_, 100 mM NaCl and 30 µM GDP) and 0.08 nM [^35^S]GTPγS (specific activity: 1250 Ci/mmole, Perkin Elmer, Waltham, MA, USA). Non-specific binding was determined with 100 µM of unlabeled GTPγS. The compounds were tested in both the agonist and antagonist mode. In the antagonist mode, 10^−6.8^ M of 8-OH-DPAT was used as a stimulating ligand. The final DMSO concentration in the assay was 5%. The reaction mixture was incubated for 90 min at 30 °C on an orbital shaker set at 250 rpm. The reaction mixture was then deposited under vacuum with the FilterMate Harvester^®^ (Perkin Elmer, Waltham, MA, USA) onto Unifilter^®^ GF/C Plates (Perkin Elmer, Waltham, MA, USA) presoaked with wash buffer (50 mM Tris-HCl, pH = 7.4). The wells were then rapidly washed with 2 mL of wash buffer. The filter plates were dried overnight at room temperature. Once completely dry, 35 µL of MicroScint PS (Perkin Elmer, Waltham, MA, USA) scintillation fluid was added to each well. Radioactivity was counted in a MicroBeta^2^ LumiJet scintillation counter (Perkin Elmer, Waltham, MA, USA). Data were analyzed with GraphPad Prism 5.0 software (GraphPad Software, San Diego, CA, USA, www.graphpad.com, accessed on 4 April 2012). The curves were fitted with the three-parameter non-linear regression model. Potency (EC_50_ or IC_50_ ± 95% confidence intervals) and efficacy (E_max_ ± SEM) were calculated and expressed as means from two separate experiments.

### 3.4. Computational Methods

In the computational part of this study, we used a protocol similar to our previous investigation on this topic [10,11,14] but based on recently obtained crystal structures of both 5-HT_1A_ and 5-HT_2A_ receptors. In the case of the 5-HT_1A_ receptor, we selected three crystal structures: apo-5-HT_1A_ (PDB id: 7e2x), serotonin-bound 5-HT_1A_ (PDB id: 7e2y) and aripiprazole-bound 5-HT_1A_ (PDB id: 7e2z), all complexed to a G protein [33]. In the case of the 5-HT_2A_ receptor, we selected two crystal structures: 5-HT_2A_ in complex with serotonin (PDB id: 7wc4) and 5-HT_2A_ in complex with aripiprazole (PDB id: 7voe) [34,35]. The choice of these particular structures was made on the basis of a very high similarity of compounds studies in this work to aripiprazole. We used two different docking protocols for all investigated coumarin derivatives. First, we manually superimposed all studied coumarin derivatives onto the aripiprazole poses from crystal structures of 5-HT_1A_ (7e2z) and 5-HT_2A_ (7voe) and performed a local search procedure using standard Autodock 4.2 parameters and 1000 independent hybrid genetic algorithm local search runs [36]. Second, we performed standard flexible docking with the Lamarckian genetic algorithm and 200 runs for each ligand–receptor pair for each of the five GPCR crystal structures. In the case of the 5-HT_1A_ receptor, the following residues are described in a flexible manner: Y96, Q97, F112, D116, T121, S199, F361, N386, and Y390, while for 5-HT_2A_ receptor flexible residues were: W151, D155, V156, F243, F332, W336, F339, F340, N363, and V366. In each case, we used 60 × 60 × 60 Å^3^ boxes centered on binding pockets of studied receptors. Additionally, we performed computational assessment of ADME properties using the QikProp 4.6 software and evaluated pKa values of basic nitrogen-containing functional groups using Epik 5.3 software [37].

## 4. Conclusions

Our studies on determining the influence of the acetyl group position in the coumarin ring on the affinity for the 5-HT_1A_ and 5-HT_2A_ receptors allowed us to draw clear and interesting conclusions regarding the structure–activity relationship for the new subfamily of coumarin derivatives selectively targeting the 5-HT_1A_ receptor. Previously published compounds containing an acetyl group in position C-8 of the coumarin ring showed, in general, greater affinities for both 5-HT receptor types. On the other hand, some newly synthesized 6-acetyl-7-hydroxy-4-methylcoumarins showed subnanomolar 5-HT_1A_ receptor affinity and potent antagonistic or agonistic properties. We previously showed that in a very similar subfamily of 8-acetyl-7-hydroxy-4-methylcoumarins, where most of the compounds studied were, analogs of the ligands described in this study showed antagonistic properties [9]. Moreover, very small structural changes, e.g., between compounds **1** and **2**, may result in different functional properties of ligands despite similar affinities. Finally, we showed that molecular docking to the recently solved crystal structures of 5-HT receptors could be a good preliminary indicator for estimating 5-HT_1A_ receptor affinity, but fails in the accurate estimation of 5-HT_2A_ affinity. 

## Data Availability

Data are contained within the article.

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
