# Peer review of "New Piperazine Derivatives of 6-Acetyl-7-hydroxy-4-methylcoumarin as 5-HT1A Receptor Agents"

_ijms, 2023, doi:10.3390/ijms24032779_

Round 1
Reviewer 1 Report (Previous Reviewer 3)
Dear authors,
Congratulations for your article. Some errors must be corrected.
In line 78, number of compounds 1-15 must be in bold.
In scheme 1: it would be great if you can indicate in the molecules where R group is attached (in the piperazine moieties), and I strongly recommend that you indicate the yields in brackets after the number of compounds.
Honestly, I think Table 1 must be presented as supplementary information, but not in the main text of the manuscript.
In line 198, when you write Table 3,it should be typed Table 4.
In line 267, you say correlations coefficients R2 = 0.58 y 0.66. Does this factor related to directly proportional correlation? If so, Can you really considerate that factors as moderate correlation?
Sincerely yours
Author Response
Reviewer 1
In line 78, number of compounds 1-15 must be in bold.
The numbers of compound are now in bold.
In scheme 1: it would be great if you can indicate in the molecules where R group is attached (in the piperazine moieties), and I strongly recommend that you indicate the yields in brackets after the number of compounds.
We have indicated in the molecules where R group is attached (in the piperazine moieties), and indicated the yields in brackets after the number of compounds.
Honestly, I think Table 1 must be presented as supplementary information, but not in the main text of the manuscript.
We believe that X-Ray Crystallography section is an important part of the results and the main text. Therefore, we have decided not to transfer Table 1 to the supplementary information.
In line 198, when you write Table 3, it should be typed Table 4.
We have corrected this mistake.
In line 267, you say correlations coefficients R2 = 0.58 y 0.66. Does this factor related to directly proportional correlation? If so, Can you really considerate that factors as moderate correlation?
Yes, this relates to a directly proportional correlation (lower i.e. “better” Ki values correlate with lower i.e. “better” IC50 and EC50 values. TO make it clear we changed into “positive correlation”. According to the most common definitions taken from the study of Evans (10.2466/pms.1996.82.3.988) values of correlation coefficient in the range of 0.4 to 0.8 can be described as moderate to strong, therefore we elected such wording.
Reviewer 2 Report (Previous Reviewer 2)
Dear colleagues, thanking you for your response to the previous comment, I would like to say that I think the paper has improved a lot. However, there are some details that can still be improved, so I suggest the following revisions.
1. Figure 1 could be presented as a table, since being the figure of a table it seemed untidy or copied.
2. Figure 4 could specify the interaction that is being represented.
3. Regarding the sentence in line 203. In my opinion it is not convincing to publish a paper that does not justify the inconsistency between calculations and experiment. The methodology should be adapted to obtain a better approach, or otherwise give a justified answer. But this sentence is very ambiguous.
4. Take care in writing and style to avoid details such as those in lines 71,87,148,150,182,197. I am referring especially to verb tenses, misspelled words, spaces, etc.
Author Response
Reviewer 2
Dear colleagues, thanking you for your response to the previous comment, I would like to say that I think the paper has improved a lot. However, there are some details that can still be improved, so I suggest the following revisions.
Figure 1 could be presented as a table, since being the figure of a table it seemed untidy or copied.
We still have not changed Figure 1 to Table 1. As we say before, we treat Figure 1 as a figure because it relates to previously published results. Only new results are presented in the tables in this manuscript. We believe that this does not affect the publication as a whole.
Figure 4 could specify the interaction that is being represented.
We have added the description of the depicted interactions in the caption to this figure.
Regarding the sentence in line 203. In my opinion it is not convincing to publish a paper that does not justify the inconsistency between calculations and experiment. The methodology should be adapted to obtain a better approach, or otherwise give a justified answer. But this sentence is very ambiguous.
Given the accuracy of the docking protocol and the methods used in similar studies such discrepancies between experimental and computational results are very common. Here since we used the crystal structure of 5HT2A receptor all errors connected to the faulty model of the receptor are minimized, therefore it is difficult to pinpoint the true problem with getting accurate values. We did, however, include some additional explanations concerning these results, but also would like to point that the calculations have only a supporting role in this mostly experimental work and additional computational studies won’t add anything to the main results of this study, which are the subnanomolar, experimental affinities of the newly designed compounds. As stated in the manuscript the computational part was performed to get an idea if it’s worth synthesizing and testing these compounds biologically.
Take care in writing and style to avoid details such as those in lines 71,87,148,150,182,197. I am referring especially to verb tenses, misspelled words, spaces, etc.
We believe that we have corrected all mistakes.
This manuscript is a resubmission of an earlier submission. The following is a list of the peer review reports and author responses from that submission.
Round 1
Reviewer 1 Report
The manuscript has too many typographical errors, there are misquoted references. The authors must correct the manuscript. Is the biological evaluation mentioned in the manuscript? All my observations are indicated in the attached PDF file.

Reviewer 2 Report
The idea of computational calculations is to complement the experiments and in this case it is not fulfilled, rather it raises questions without proper answers. In the abstract it was mentioned that it was used to design compounds but this is not observed in the work methodology nor results. Docking, in general is not the right tool to predict affinity comparable to experimental Ki, however it could be used to complement the discussion of possible interactions according to ligand structure but it should be done in a cleaner way. Check the dimensions of the box as they seem to be exceeded.
Reviewer 3 Report
Dear authors,
It is good to know they could synthesize and very well characterized the new piperazine derivatives, which represent an interesting extension of your previous work.
However, major considerations might to be corrected in the text and in the way of presentation. Just few examples:
1- Figure 1 should be rename Table 1. (Line 52)
2. It is very difficult to read the full names of all compounds (from line 80 to 85). And they again appear along to the text.
3. X- Ray Crystalography should be deleted from the manuscript and it should be place in supplementary information.
4. Table 5 is not mentioned in the text.
5. Line 211, compound 2, and line 225 (the numbers of compounds should be bold)
6. Can you explain what is WAY 100636 (line 237) ?
In the part of characterization of products, several considerations:
1. Which amounts do you use for reactions? Which scale?
2. In spite of you stablish the eluent of TLC in the text, you should write it in line 289. Is always the same eluent for all products?
3. Line 302: At 6.98 ppm you say correspond with H-2´´-... You should type H-3´´.....
4. Line 312: At 7.20 ppm, you say H-5´´. Impossible, you should type H-3´´.
5. Line 349: It is missing the carbon at 69.3 ppm
6. Line 386: Where you type 199.8 ppm, you should write 197.8 ppm, according to real spectra provided in supporting information
In general, you should put attention to the structure of the text, specially the headboards. For expample, Conclussions appear duplicated and numeration of subtittles are uncorrect.
Sincerely yours